# Chemical Profile of the Volatile Constituents and Antimicrobial Activity of the Essential Oils from *Croton adipatus*, *Croton thurifer,* and *Croton collinus*

**DOI:** 10.3390/antibiotics10111387

**Published:** 2021-11-12

**Authors:** Juana Liz Leslie Cucho-Medrano, Sammy Wesley Mendoza-Beingolea, César Máximo Fuertes-Ruitón, María Elena Salazar-Salvatierra, Oscar Herrera-Calderon

**Affiliations:** 1Faculty of Pharmacy and Biochemistry, Universidad Nacional Mayor de San Marcos, Jr. Puno 1002, Lima 15001, Peru; liz.cumed@gmail.com (J.L.L.C.-M.); sammy.mendoza@unmsm.edu.pe (S.W.M.-B.); 2Institute for Research in Pharmaceutical Sciences and Natural Resources, Faculty of Pharmacy and Biochemistry, Universidad Nacional Mayor de San Marcos, Jr. Puno 1002, Lima 15001, Peru; cfuertesr@unmsm.edu.pe; 3Research Institute in Biological Chemistry, Microbiology and Biotechnology, Faculty of Pharmacy and Biochemistry, Universidad Nacional Mayor de San Marcos, Jr. Puno 1002, Lima 15001, Peru; msalazars@unmsm.edu.pe; 4Department of Pharmacology, Bromatology and Toxicology, Faculty of Pharmacy and Biochemistry, Universidad Nacional Mayor de San Marcos, Jr. Puno 1002, Lima 15001, Peru

**Keywords:** antimicrobial, antifungal, colorimetric microdilution, volatile oil, medicinal plants, gram positive, gram negative

## Abstract

The aim of this study was to determine the volatile phytochemical constituents and evaluate the antimicrobial activity of the essential oils of the leaves from *Croton adipatus*, *Croton thurifer*, and *Croton collinus*. Essential oils were extracted by hydro-distillation using the Clevenger extractor and the phytochemical analysis was determined by Gas chromatography-Mass Spectrometry (GC-MS). The antimicrobial activity was assessed using the agar diffusion and colorimetric broth microdilution methods against *Staphylococcus aureus* ATCC 25923, *Bacillus subtilis* ATCC 6633, *Escherichia coli* ATCC 25922, *Pseudomonas aeruginosa* ATCC 9027, and *Candida albicans* ATCC The essential oils from *C. adipatus*, *C. thurifer*, and *C collinus* had 46, 38, and 35 volatile constituents respectively. The main compounds determined in *C. adipatus* were β-myrcene (18.34%), while in *C. collinus* was β-caryophyllene (44.7%), and in *C. thurifer* was an unknown component (C_10_H_16_: 22.38%). Essential oil of *C. adipatus* showed a MIC against *B. subtilis* (286.4 µg/mL) and *C. albicans* (572.8 ± 0 µg/mL); *C. thurifer* against *S. aureus* (296.1 ± 0 µg/mL) and *B. subtilis* (148 ± 0 µg/mL); and *C. collinus* against *B. subtilis* (72 ± 0 µg/mL) and *C. albicans* (576.2 ± 0 µg/mL). The three essential oils of Croton species demonstrated *in vitro* antimicrobial activity against a strain of bacteria or fungi.

## 1. Introduction

In recent decades, the emergence of antimicrobial resistance has generated alarm in the health system. Several factors have contributed to the evolution of antimicrobial resistance, including microbial adaptation through selective biochemical processes, self-medication in the community, and the shortage in the development of new antimicrobial molecules which leads to choices of antimicrobial drugs with an inadequately broad spectrum and side effects, making access to essential antimicrobials more expensive [1]. On the other hand, the medicinal properties of plants have been recognized and have generated a great deal of interest due to their easy access, low toxicity, economic viability and easy preparation forms such as macerations, decoctions, infusions, tinctures, and essential oils [2]. Hence, there are many alternative therapies that include the extended use of essential oils, which are considered a set of volatile secondary metabolites synthesized in different plant organs as a response to adverse environmental conditions, and the attack of herbivores [3]. Several research studies attribute different medicinal properties to essential oils such as antibacterial, antifungal, antiviral, antioxidant, anti-inflammatory, anticancer, antidiabetic, antiparasitic, insecticidal, antimutagenic, antiprotozoal, immunomodulatory, and antitumor [4].

Peru is a megadiverse country due to its geographical location; the Peruvian flora comprises around 25,000 species that are distributed in the different ecological levels. An important part of the flora grows in the inter-Andean valleys and are exposed to a high solar radiation and low temperatures [5]. Among the plants belonging to the Euphorbiaceae family, Peru presents 61 genera, and 323 species of which the Croton genus is the most numerous in endemic species [6]. Additionally, Croton is the second most abundant genus of the Euphorbiaceae family due to its variety and richness, which is why it includes more than 1200 species. The Croton genus presents a cosmopolitan distribution both in the tropics and subtropics throughout the world, primarily in arid and semi-arid zones [7,8]; however, a wide variety are found in Madagascar, Brazil, and the Caribbean [9]. In Peru they are found in different regions between altitudes ranging from 110 to 3200 meters above sea level.

The Croton genus is important locally for its different medicinal properties, i.e., their leaves are used as healing, hemostatic, antidiarrheal, anti-inflammatory, and antiseptic agents. They are used for conditions such as uterine colic, gastrointestinal ulcers, urinary retention, antepartum vaginal baths, postpartum hemorrhages, skin diseases, and for the control of fever associated with gastrointestinal illnesses [10]. Essential oils from Croton species have presented several volatile constituents as fingerprint, which might be related to its antimicrobial activity, i.e., *C. ceanothifolius* showed bicyclogermacrene (26.3%), germacrene D (14.7%), and E-caryophyllene (11.7%) as major components [11], in *C. heliotropiifolius* were found β-caryophyllene as the major constituent followed by bicyclogermacrene, germacrene-D, limonene, and 1,8-cineole [12], in *C. argyrophyllus* had bicyclogermacrene (38.42 %), (Z)-caryophyllene (14.06 %), epi-longipinanol (9.78 %), and germacrene B (9.1 %) [13], in *C. zenhtneri, C. nepetaefolius,* and *C. sonderianus* were trans-anethole (94.09%), methyl eugenol (48.47%), and β-phellandrene (18.21%) respectively [14], finally in *C. pulegiodorus* were identified β-caryophyllene (20.96%), bicyclogermacrene (16.89%), germacrene-D (10.55%), τ-cadinol (4.56%), and β-copaen-4-α-ol (4.35%) [15].

Otherwise, *Croton adipatus*, *Croton thurifer*, and *Croton collinus* are known for their antimicrobial properties; however, such properties have not been scientifically proven as well as their chemical composition. Thus, the main aim in this study was to determine the chemical composition and the *in-vitro* antimicrobial activity of the essential oils from *Croton adipatus*, *Croton thurifer*, and *Croton collinus* against *Staphylococcus aureus* ATCC 25923, *Bacillus subtilis* ATCC 6633, *Escherichia coli* ATCC 25922, *Pseudomonas aeruginosa* ATCC 9027, and *Candida albicans* ATCC 10231.

## 2. Results and Discussion

### 2.1. Phytochemical Screening of the Volatile Components of the Essential Oils from Croton Species

The extraction yields of the essential oils of *Croton adipatus*, *Croton thurifer*, and *Croton collinus* were 0.47 ± 0.01%, 0.07 ± 0.005%, and 0.06 ± 0.001%, respectively; being the first reports in these species (Appendix A). Other studies reported yield percentages in *C. borarium* leaves and *C. geayi* leaves values of 0.68% and 0.72%, respectively [16], *C. zambesicus* (leaves: 0.29%) [17], *C. cajucara* [18](leaves: 0.4% and 0.65%), *C. heterocalyx* (leaves: 0.45%) [19], *C. pullei* (stems: 0.06%, leaves: 0.5%) [20], *C. campestris* (branches: 0.02%; leaves: 0.04%) [21] *C. blanchetianus* (leaves: 0.7%) [22], *C. oblongifolius* (bark: 0.9%) [23], and *C. hieronymi* (roots: 0.06%; leaves: 0.07%) [24]. In this study, *C. adipatus, C. thurifer, and C. collinus* showed extraction yields similar to the average reported in other species of the Croton genus. Several factors such as soil composition, climate, geographic location, variety of the species, vegetative state, seasonal variation, organ used for extraction, collection period, storage, drying methods, extraction processes, and others might be decisive in the composition, proportion, and quantity in the yield rate of essential oils [25].

Regarding to the determination of the volatile phytochemicals by GC-MS of the essential oil of *C. adipatus*, 46 compounds were determined (Table 1; Appendix A, Appendix A), of which monoterpenes represent 72.73% (monoterpenes hydrocarbon 69.76%; oxygenated monoterpenes 2.97%), followed by sesquiterpenes 18.82% (sesquiterpenes hydrocarbon 16.06%; oxygenated sesquiterpenes 2.76%), phenylpropane derivatives 0.24%, and others 8.21%. β-myrcene was the most abundant component with 18.34% followed by α-thujene at 12.69%, D-limonene at 10.94%, α-phellandrene at 8.19%, and β-elemene at 6.47%. These results are the first report of the chemical composition; the concentrations of β-myrcene (18.34%) and α-thujene (12.69%) are the major ones found in regard to other species of the Croton genus. A similar concentration of D-limonene (10.94%) was also found by De Almeida et al. in *C. campestris* (9.7%) [21], and a lower concentration in *C. geayi* (22.92%) [16].

For the essential oil of *C. thurifer*, 35 compounds were determined (Table 1; Appendix A, Appendix A). The composition of sesquiterpenes were predominantly abundant, at 62.26% (sesquiterpenes hydrocarbon 33.88%; oxygenated sesquiterpenes 28.38%), followed by monoterpenes at 35.39% (monoterpenes hydrocarbon 34.88%; oxygenated monoterpenes 0.51%) and other at 2.35%. The unknown monoterpenes component with 22.38% (C_10_H_16_; Rt: 18.57) was the most abundant component, followed by another unknown sesquiterpene component 21.8% (C_15_H_26_O; Rt: 53.7), β-elemene 11.87%, germacrene D 10.22%. Moreno et al. reported in *C. heterocalyx* leaves, β-elemene (8.2%), germacrene D (12.5%) [19] and De Araújo reported *C. rhamnifolius* stems (17.28%) [26]. On the other hand, Dai et al. in *C. cascarilloides* (6.0%) [27] and Turiel et al. in *C. draconoides* (9.0%) [28].

In *C. collinus* essential oil, 38 compounds were determined (Table 1; Appendix A, Appendix A). Sesquiterpenes 70.26% were obtained predominantly (hydrocarbon sesquiterpenes 68.25%; oxygenated sesquiterpenes 2.51%), followed by monoterpenes 28.03% (hydrocarbon monoterpenes 27.44%; oxygenated monoterpenes 0.59%) and others 1.21%. The major component was β-caryophyllene with 44.7%, followed by D-limonene 8.73%, β-thujene 6.96%, β-myrcene 6.79%, and β-elemene 6.7%. β-caryophyllene was the highest abundant volatile component among Croton species. However, other studies reported lower concentrations of this component (β-caryophyllene): in *C. antanosiensis* leaves (28.23%), *C. decaryi* leaves (26.65%) [29], *C. rhamnifolioides* (6.33%) [30], *C. huberi* (18.3%) [31], *C. conduplicatus* (7.8%) [32], *C. trinitatis* (15.3%) [33], *C. isabelli* (14.3%), and *C. pallidulus* (11.5%) [34].

Other studies of the Croton genus reported other major compounds according to the organ of the plant such as in *C. zambesicus* leaves and stems (1,8-cineole 27.07% and cymene 13.80%) [35]; *C. zehntneri* leaves (estragole 93.61% and 84.7%) [36], *C. hieronymi* leaves (camphor 13.9%) [24], *C. hieronymi* root (γ-asarone 25.7%) [24], *C. oblongifolius* bark (terpinen-4-ol 17.8%) [23]; *C. pullei* leaves (linalol 24.9%) [20], *C. greveanus* leaves (1.8 cineol 40.40% and linalol 23.81%); *C. borarium* leaves (β-phellandrene 39. 72% and α-terpineol 25.12%); and in *C. geayi* leaves (β-pinene 28.74% and limonene 22.92%) [16]. The diversity of the qualitative and quantitative characteristics of the components of essential oils in *C. adipatus*, *C. thurifer*, and *C. collinus* can be explained due to many factors reported by other authors such as the harvesting season, extraction method, drying method, drying time, geographical conditions, genetic variability, soil composition, plant organ, climate, drought condition, vegetative state cycle, plant nutrition, application of fertilizers, stress during growth, and post-harvest storage [25,37,38].

On the other hand, differences in the qualitative and quantitative compositions of the obtained essential oils have been observed in Table As shown in the Venn diagram (Figure 1), α-thujene, α-pinene, sabinene, β-pinene, α-phellandrene, D-limonene, β-elemene, β-caryophyllene, germacrene D, α-cadinene, and heptadecane were shared by all three species, for a total of 11 compounds. Additionally, 18 compounds were uniquely identified in *C. adipatus*, 11, and 10 in *C thurifer* and *C. collinus*, respectively. 

### 2.2. Antimicrobial Activity of the Essential Oils of Croton Species

#### 2.2.1. Agar Diffusion Method

Table 2 shows that *C. adipatus* essential oil at a concentration of 50% inhibited the growth of *S aureus* (20.0 ± 0 mm), while a concentration of 95% inhibited the growth of *B. subtilis* (25.0 ± 0 mm) and *E. coli* (20.03 ± 0.29 mm), while at 10% inhibited *C. albicans* (19.67 ± 1.47 mm). On the other hand, no significant activity was observed against *P. aeruginosa*. Additionally, *C. thurifer* (concentration 10%, 50% and 95%) essential oil presented inhibition halos ranging from 11 to 14 mm for *S. aureus*, *B. subtilis*, *E. coli*, and *P. aeruginosa*. In *C. albicans* was observed an inhibition halo of 15.03 ± 0.38 mm at 10%. *C. collinus* (concentration 10%, 50% and 95%) presented inhibition halos between 11 and 15 mm for *S. aureus*, *B. subtilis*, *E. coli*, and *C. albicans* (Table 2 and Figure 2). 

In other studies of the Croton genus regarding the antimicrobial activity (diffusion method, the following inhibition halos were reported:

In *S. aureus*: *C. borarium* leaves at 100% (22.5 mm) [16] and *C. gratissimus* leaves at 5 mg/mL (21.6 mm) [39]. Similar results were obtained in our study of *C. adipatus* at 50% (20.0 mm). On the other hand, other authors reported that the essential oils of the following plant species did not have good activity: leaves and stems of *C. zehntneri* at 50 µg/disc (14.53 ± 0.25 mm) [36] in *C. geayi* leaves at 100% (10 mm) and *C. greveanus* leaves *at* 100% (11.5 mm) [16]; as well as in *C. thurifer* at 95% (14.07 mm) and *C. collinus* at 95% (12 mm) (Table 2 and Figure 2).

Regarding *B. subtilis;* the essential oils of *C. zehntneri* at 5 µg/disc and 25 µg/disc (<8 mm and 13.33 ± 0.42 mm, respectively) [36], *C. borarium* (15 mm), *C. geayi* (9 mm), and *C. greveanus* leaves (13.2 mm) at 100% [16] did not have a significant activity against this strain. Similar results were evidenced in *C. thurifer* at 50% (12.53 mm) and *C. collinus at 95%* (15 mm). However, *C. adipatus* at 95% showed an inhibition halo of 25.0 mm (Table 2 and Figure 2). 

For *E. coli*, the essential oil from *C. gratissimus* leaves at 5 mg/mL reported an inhibition halo of 23.0 mm [39], similar results were obtained with *C. adipatus* at 95% (20.03 mm). On the other hand, other studies reported that the essential oils of the following plant species did not have good activity: *C. zehntneri* leaves and stems [36], *C. hieronymi* roots and leaves [24], *C. borarium*, *C. geayi,* and *C. greveanus* leaves [16]; similar results were observed in *C. thurifer* at 50% (12.03 mm) and *C. collinus* at 50% (12.7 mm) (Table 2 and Figure 2). 

In *P. aeruginosa*, *C. borarium* leaves at 100% showed an inhibition halo of 18.3 mm [16]. In contrast, *C. zehntneri* at 5, 25, and 50 ug/disc (<8 mm) [36], *C. geayi* leaves at 100 % (3.17 mm), *C. greveanus* leaves at 100% (8.01 mm) [16], and *C. gratissimus* leaves at 5 mg/mL (11.0 ± 0.6 mm) [39] did not show antimicrobial activity being similar with our findings, which ranges between 10 ± 0 mm and 13.57 ± 0.85 mm (Table 2 and Figure 2). 

For *C. albicans*, the following Croton species did not have a significant activity such as *C. zehntneri* leaves and stems at 5 ug/disc (<8 mm) and 25ug/disc (<8 mm) [36], *C. hieronymi* roots and leaves at 1000 µg/mL (7.9 and 0 mm, respectively) [24], and *C. cajucara* at 100% had ranging from 6–10 mm [18]. Similar results were obtained for *C. thurifer* and *C. collinus* with inhibition zones between 10 and 15 mm. Otherwise, in the present study, *C. adipatus* at 10% reported an inhibition halo of 19.67 mm (Table 2 and Figure 2). 

Results for this type of preliminary antimicrobial test showed a certain degree of variability, which could be explained by different factors that influence the solubility of the essential oils in the medium to be diffused, the polarity of the essential oil, and the molecular size of the components that comprise the essential oil, among others; therefore, they would not necessarily correlate with the results of other specific tests such as microdilution or dilution in broth. Regarding the biochemical mechanisms involved in the antimicrobial activity of essential oils, some compounds like p-cymene at 2500 µg/mL have shown alterations in the membrane potential of *S. aureus*, and also γ-terpinene at 2500 and 3400 µg/mL [40]. β-caryophyllene, the major component in *C. collinus* was able to alter membrane permeability and integrity of *B. cereus*, leading possibly to cell death [41]. This finding is related to our results against *B. subtilis*, which only was selective for this strain (Table 2 and Table 3). Limonene, the third abundant component in *C. adipatus*, is linked to the respiratory metabolism of *S. aureus* [42]. Otherwise, terpenes are also related to disorders of mitochondrial membrane in Candida species but with low antifungal activity. Moreover, oxygenated sesquiterpenes demonstrated a negative effect on Saccharomyces and Candida strains such as caryophyllene oxide, humulene epoxide, spathulenol, and α-Muurolol in high concentrations [43].

#### 2.2.2. Colorimetric Microdilution Method

According to our results showed in Table 3 and Figure 3, the essential oil of *C. adipatus* showed antimicrobial activity against *B. subtilis* (MIC = 286.4 µg/mL) and *C. albicans* (MIC = 572.8 µg/mL) and did not show activity against *S. aureus* ATCC 25923 (MIC > 1000 µg/mL), *E.*coli (MIC = 1000 µg/mL) and *P. aeruginosa* (MIC > 1000 µg/mL). The essential oil of *C. thurifer* showed activity against the following bacteria: *S. aureus* (MIC = 296.1 µg/mL) and *B. subtilis* (MIC = 148 µg/mL). Furthermore, it did not show activity against *E. coli* (MIC > 1000 µg/mL), *P. aeruginosa* (MIC > 1000 µg/mL), and *C. albicans* (MIC => 1000 µg/mL). The essential oil of *C. collinus* showed antimicrobial activity against *B. subtilis* (MIC = 72 µg/mL), and *C. albicans* (MIC = 576.2 µg/mL). Additionally, it did not show activity against *S. aureus* (MIC > 1000 µg/mL), *E. coli* (MIC > 1000 µg/mL), and *P. aeruginosa* (MIC > 1000 µg/mL) (Table 3 and Figure 3).

In *S. aureus*, good activity was evidenced in the aerial parts of *C. zambesicus* (MIC = 16 µg/mL) [17]. *C. cajucara* leaves (MIC = 33.4 µg/mL) [18], *C. campestris* (MIC = 128 µg/mL) [32], *C. zehntneri* stems and leaves (MIC = 145.0 µg/mL) [36], in the aerial parts of *C. heliotropiifolius* (MIC = 500 µg/mL) [44], *C. limae* leaves (MIC = 512 µg/mL) [45], and *C. gratissimus* leaves (MIC = 600 µg/mL) [39]. Similar results were obtained in the present study for *C. thurifer* leaves (MIC = 296.1 µg/mL). On the other hand, other investigations reported that the essential oils of the following plant species did not have a good activity such as *C. borarium* (MIC = 1151 µg/mL) [16], *C. greveanus* (MIC = 1126 µg/mL) [16], and *C. geayi* (MIC = 2260 µg/mL) [16]. 

In *B. subtilis*, other Croton species presented good activity such as: *C. zambesicus* steam and leaves (MIC = 16 µg/mL) [17], *C. heliotropiifolius* aerial parts (MIC = 62.5 µg/mL) [44], *C. zehntneri* steam and leaves (MIC = 58.75 µg/mL) [36], and *C. borarium* leaves (MIC = 287 µg/mL) [16]. Similar results were obtained for the essential oils of the leaves of the three species of Croton, obtaining better results for *C. collinus* (MIC = 72 µg/mL), followed by *C. thurifer* (MIC = 148 µg/mL), and *C. adipatus* (MIC = 286.4 µg/mL). On the other hand, *C. greveanus* leaves (MIC = 1126 µg/mL) [16], and *C. geayi* (MIC = 4520 µg/mL) [16] did not present activity against this microorganism. 

For *E. coli*, the following authors reported a significant activity in *C. zambesicus* stems and leaves (MIC = 16 µg/mL) [17] and *C. hieronymi* (MIC = 100 µg/mL) [24]. On the other hand, *C. limae* leaves (MIC ≥ 1024 µg/mL) [45], *C. boraium* leaves (MIC = 1151 µg/mL) [45], *C. greveanus* (MIC = 4505 µg/mL) [16], and *C. gratissimus* leaves (MIC = 1300 µg/mL) [16]. Similarly, in our study, no activity was found in *C. adipatus*, *C. thurifer* and *C. collinus*. 

In *P. aeruginosa*, *C. zambesicus* leaves showed an inhibition at 250 µg/mL [17]. Meanwhile *C. limae* leaves (MIC = 1024 µg/mL) [17], *C. boraium* leaves (MIC= 1151 µg/mL) [16], *C. greveanus* (MIC = 4505 µg/mL) [16], and *C. gratissimus* leaves (MIC=5000 µg/mL) [39] did not show promising results, as well as the studied species of Croton in our study. 

For *C. albicans*, a good activity was reported according to Alviano et al. in *C. cajucara* leaves (MIC = 13.4 µg/mL) [18], *C. zehntneri* stem and leaves (MIC = 58.75 µg/mL) [36], and in *C. hieronymi* roots (MIC = 100 µg/mL) [24]. Similar results were presented in the present study for the species of *C. adipatus* (MIC = 571 µg/mL) and *C. collinus* (MIC = 574 µg/mL). On the other hand, *C. hieronymi* roots (MIC > 1000 µg/mL) [24] did not present activity, in the same way with *C. thurifer* (MIC > 1000 µg/mL). MIC of standard drugs are shown in Table 3 and in Appendix A.

As is evidenced in the present study, the species of *C. adipatus, C. thurifer,* and *C. collinus* showed antimicrobial activity according to the microdilution test against Gram-positive and yeast. Other studies of the Croton species revealed diverse activity on the microorganisms under study, which means the high variability of the components present and associated with this activity. Various studies mention a slight sensitivity of Gram positive compared to Gram-negative. The main reason is the presence of the extracellular membrane in Gram-negative microorganisms, which regulate the passage of hydrophobic components and thus prevents the alteration of the membrane’s permeability [46]. Likewise, various studies attribute the antimicrobial activity in essential oils to the following components: terpinen-4-ol [4], α-terpinene [47], 1,8-cineole [48], δ-cadinene, γ-muurolene, α-muurolene [49], β-caryophyllene [50], β-elemene [51], germacrene [52], p-cymene [53], γ-terpinene [54], β-myrcene [55], α-pinene [56], β-pinene [56], linalool [57], α-phellandrene [58], among others. All these components were found in the essential oils of Croton in our study. Furthermore, the observed activity could also be explained by the presence of the components in higher proportion or by the synergistic action between them.

Based on our results, we might suppose that the three essential oils had a selective effect against a specific bacterium and on *C. albicans*. Each metabolite determined could be acting by different mechanisms and according to our chemical analysis, each Croton species showed a different phytochemical marker, which contributes to the antimicrobial activity. Hydrocarbon monoterpenes were the most abundant chemical groups in *C. adipatus*. However, in *C. thurifer* and *C. collinus* were the sesquiterpenes, which might be linked to this difference in the results showed in Table 2 and Table Therefore, the volatile chemicals would be responsible for the antimicrobial effect on *S. aureus*, *B. subtilis*, and *C. albicans,* and not the major phytochemical determined in the GC-MS analysis. Some studies reported that terpenes do not constitute a high inherent antimicrobial effect i.e., some terpenes like p-cymene, the major compound in thyme, had no effect against several Gram-negative bacteria. Also, other chemicals, which were found in Croton essential oils such as α-pinene, limonene, δ-3-carene, sabinene, α-pinene, β-pinene, and α-terpinene revealed no or low effect against different pathogen bacteria, whether Gram-negative or Gram--positive. Additionally, α-pinene, β-pinene, p-cymene, β-myrcene, β-caryophyllene, limonene, and γ-terpinene had low or absent activity against *Escherichia coli, Staphylococcus aureus*, and *Bacillus cereus* and p-cymene and γ-terpinene were ineffective against *Saccharomyces cerevisiae* [40]. Thus, these reports in vitro indicate that single terpene compounds are inefficient as antimicrobials, which supports our study to understand the relationship between the chemical composition of each Croton species with antimicrobial activity. 

## 3. Materials and Methods

### 3.1. Collection of the Plant Material

*Croton adipatus* Kunth and *Croton thurifer* Kunth were collected in the locality El Almendral in the district of Jaen of the province of Jaen, Departament of Cajamarca, at an altitude of 650 meters above sea level, and *Croton collinus* Kunth was collected in Jahuanga, district of Bagua Grande in the province of Utcubamba of the department of Amazonas, at an altitude of 700 meters above sea level, in June 2017 during the dry season. Leaves were separated, selected and dried in open air, avoiding exposure to direct sunlight in order to preserve the volatile components and also to achieve a longer storage time. The identification and taxonomic classification of plant species were developed by the botanical consultant José Campos de la Cruz.

### 3.2. Extraction of Essential Oils of Croton Species

The extraction was developed by hydrodistillation using a Clevenger apparatus for two hours. Milligrams of anhydrous sulfate were added to remove residual water. Then it was stored at 4°C until further use [59]. The following formula was used to calculate the yield percentage:$$Yield\;percentage\;\left(\%\right)\;=\frac{volume\;of\;EO\;in\;mL\;}{Weight\;of\;EO\;in\;g}\;\times 100$$
where *EO*: essential oil.

### 3.3. Chemical Analysis by Gas Chromatography Coupled to Mass Spectrometry (GC-MS) of the Essential Oils of Croton Species

For the analysis of each Croton species, 20 µL of the essential oil was used in 980 µL of dichloromethane, which were injected into the Agilent Technologies 7890A gas chromatograph coupled to an Agilent Technologies 5975C mass selective detector. The separation of the compounds from the mixture was carried out using a DB-5MS apolar capillary column (60 m × 250 µm × 0.25 µm) (J&W of 5% phenyl methylpolysiloxane). The injector temperature was kept at 250 °C and the injection was carried out in split mode (20: 1). The oven temperature program was as follows: initial temperature 50 ° C; subsequently, it was increased to 2.5 °C/min up to 180 °C, 10 °C/min up to 200 °C, and finally at 20 °C/min up to 240 °C. The run time was of 56 min, using helium as a stripping gas at a constant flow of 1mL/min. The constituents of essential oils were identified by comparing the mass spectra of each peak with those of the mass spectra library of the Flavor 2 databases and that of the National Institute of Standards and Technology (NIST, 08). This analysis was carried out in the laboratory of the Natural Products Research Unit of the Faculty of Sciences and Philosophy "Alberto Cazorla Talleri", Universidad Peruana Cayetano Heredia.

### 3.4. Evaluation of the Antimicrobial Activity of the Essential Oils of Croton Species

#### 3.4.1. Agar Diffusion Method

The antibacterial activity of C. adipatus, C. thurifer, and C. collinus essential oils were tested against the following species: Staphylococcus aureus ATCC 25923, Bacillus subtilis ATCC 6633, Escherichia coli ATCC 25922, Pseudomonas aeruginosa ATCC 9027 and the yeast Candida albicans ATCC Bacteria and yeast were sub-cultured on Trypticase Soy agar and Dextrose Sabouraud agar, respectively, 24 h before the test. The suspension of the microorganisms was prepared in sterile 0.9% saline solution, and the turbidity was adjusted according to the 0.5 tube of the McFarland scale, which corresponds to an approximate concentration of 1–2 × 10^8^ CFU/mL for bacteria and 1–5 × 10^6^ CFU/mL for yeast. 

Petri dishes were prepared as follows: Müeller Hinton agar was used for bacteria and Sabouraud Dextrose agar for yeast, previously reconstituted with distilled water, sterilized by autoclaving, cooled and kept between 45 °C and 50 °C. These were inoculated with 0.4 mL of inoculum suspension for every 100 mL of culture medium, then it was homogenized and distributed in sterile glass Petri dishes of 90 mm in diameter, at a rate of 25 mL per plate. It was allowed to solidify, and wells were made using a sterile cork-borer. Then, 100 μL of the concentrations of the essential oils (10%, 50%, and 95%) were incorporated in each well, and finally were kept for 30 min under refrigeration to enhance the diffusion of the essential oils and to avoid the proliferation of microorganisms, then it was incubated in an oven at 37 °C for 24 h. Ciprofloxacin (0.05 mg/mL) and Ketoconazole (0.2 mg/mL) dissolved in water and dimethylsulfoxide (DMSO), respectively, were used as positive controls. Solvent controls (DMSO) were included in each experiment as negative controls. Tests were carried out in triplicate. After incubation, the presence of growth inhibition zones was observed, and the diameters were measured in mm. It was considered a significant antimicrobial activity to a perfectly clear area with a diameter greater than 18 mm [60].

#### 3.4.2. Colorimetric Microdilution Method and Determination of MIC

The antibacterial activity was performed by colorimetric microdilution method utilizing 96-well microtiter plates to determine the minimum inhibitory concentration (MIC). The following microorganisms were used: Staphylococcus aureus ATCC 25923, Bacillus subtilis ATCC 6633, Escherichia coli ATCC 25922, Pseudomonas aeruginosa ATCC 9027, and Candida albicans ATCC The microorganisms were kept for 24 h before the test on Trypticase Soya agar for bacteria and Dextrose Sabouraud agar for yeast and incubated at aerobic conditions at 37 °C. Later, a few colonies of the microorganisms in sterile 0.9% saline solution was prepared with the help of a Kolle handle and the turbidity of the bacterial suspension was adjusted according to the 0.5 tube of the McFarland scale, which corresponds to an approximate concentration of 1–2 × 10^8^ CFU/mL for bacteria and 1–5 × 10^6^ CFU/mL for yeasts. In bacteria, dilutions were made with Müeller Hinton broth to obtain an inoculum of 6.6–13.3 × 10^5^ CFU/mL (2x inoculum). For yeast, consecutive dilutions were made with RPMI 1640 medium to obtain an inoculum of 1–5 × 10^3^ CFU/mL (2X inoculum). For each 20 mL of the suspensions of the 2x inocula (bacteria or yeast), 0.1 mL of 20 mg/mL resazurin solution was added, which was previously prepared under aseptic conditions using a 0.22 µm filter. The stock solutions of the essential oils were prepared at a concentration of 800 µL/mL [essential oil (80): polysorbate 20 (50): Müeller Hinton broth (870)], for testing in bacteria and yeast. The final concentrations (x) of the samples ranged from 0.079 to 40 μL/mL. The positive controls used for the tests were Ciprofloxacin and Ketoconazole, following the CLSI recommendations [61]. The final concentrations (x) of the controls ranging from 0.125 to 64 μg/mL and from 0.0313 to 16 μg/mL for Ciprofloxacin and Ketoconazole respectively. The Müeller Hinton broth with resazurin was used as a sterility control in the case of Ciprofloxacin and the RPMI 1640 medium with resazurin according to the protocol of Liu M et al. [62] in Ketoconazole.

100 µL (essential oil or positive control) of the 2x inoculum was mixed with resazurin indicator. On the other hand, sterility control wells contained Müeller Hinton broth or RPMI 1640 medium, containing resazurin and growth control wells had Müeller Hinton broth for bacteria or RPMI 1640 for Candida albicans. At the end, 96-well microtiter plates were labeled and covered in polyethylene bags to avoid volatilization of the samples. The microtiter plates were incubated at 37 °C for 24 h under aerobic conditions. It was performed visually and any change in color from purple (resazurin) to pink resorufin was considered a positive result. MIC values are defined as the lowest concentration of essential oil that prevents a color change of resazurin. The average of three values were calculated and reported as the MIC. For the result interpretation, the parameters of Holetz et al [63] were considered as follows: Inactive > 1000 μg/mL, Weak activity 500 to 1000 μg/mL, moderate activity 100 to <500 μg/mL, good activity < 100 μg/mL.

### 3.5. Statistical Treatment

The results were subjected to an analysis of variance (ANOVA) followed by the Dunnet test. It is considered a significant difference when P values is less than 0.05.

## 4. Conclusions

Based on the obtained results, it can be concluded that the three essential oils from Croton are richest in hydrocarbons, monoterpenes, and hydrocarbons and oxygenated sesquiterpenes. In addition to the main bioactive compounds, C adipatus is rich in β-Myrcene (18.34%), C thurifer in an unknown monoterpene (C_10_H_16_) with 22.38% and C. collinus in β-caryophyllene (44.7%). Our research showed C. adipatus was selective for B. subtilis and C. albicans, C. thurifer for S. aureus. while C. collinus exhibited their antimicrobial activity towards B. subtilis and C. albicans. Although results showed some potential in the in vitro assay, these still may not be applied in vivo. Further research in in vivo models is necessary to evaluate the antimicrobial activity of these essential oils.

## Figures and Tables

**Figure 1 antibiotics-10-01387-f001:**
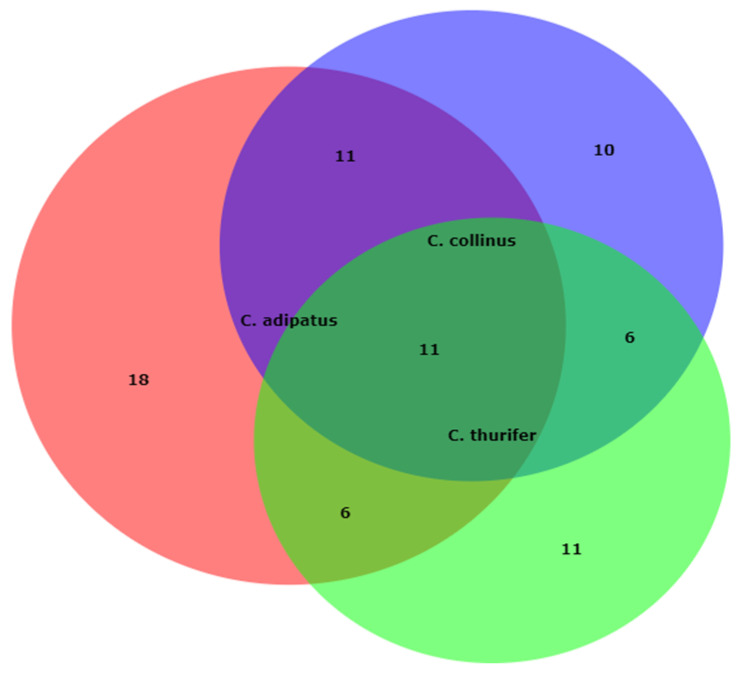
Venn diagram showing both the number of compounds shared and unshared/peculiar among the three Croton essential oils.

**Figure 2 antibiotics-10-01387-f002:**
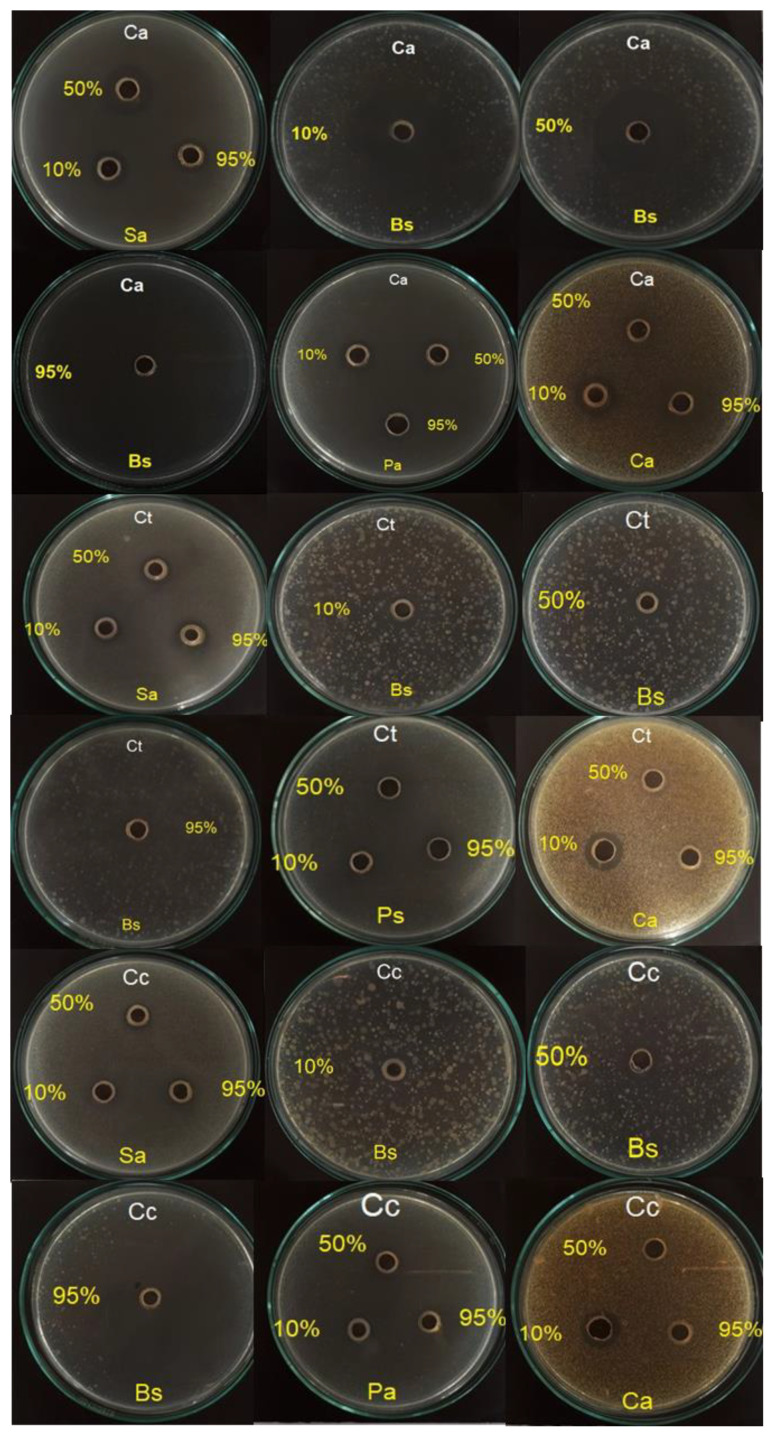
Antimicrobial activity of Croton species at different concentrations 10%, 50% and 95% by the agar diffusion method. The white acronym represents the analyzed plant species: Ca (*C. adipatus*); Ct (*C. thurifer*), Cc (*C. collinus*); while the yellow acronym represents microorganisms such as Sa (*S. aureus*), Bs (*B. subtilis*), Pa (*P. aeruginosa*), and Ca (*C. albicans*); the yellow letters represent the concentration percentages of each essential oil.

**Figure 3 antibiotics-10-01387-f003:**
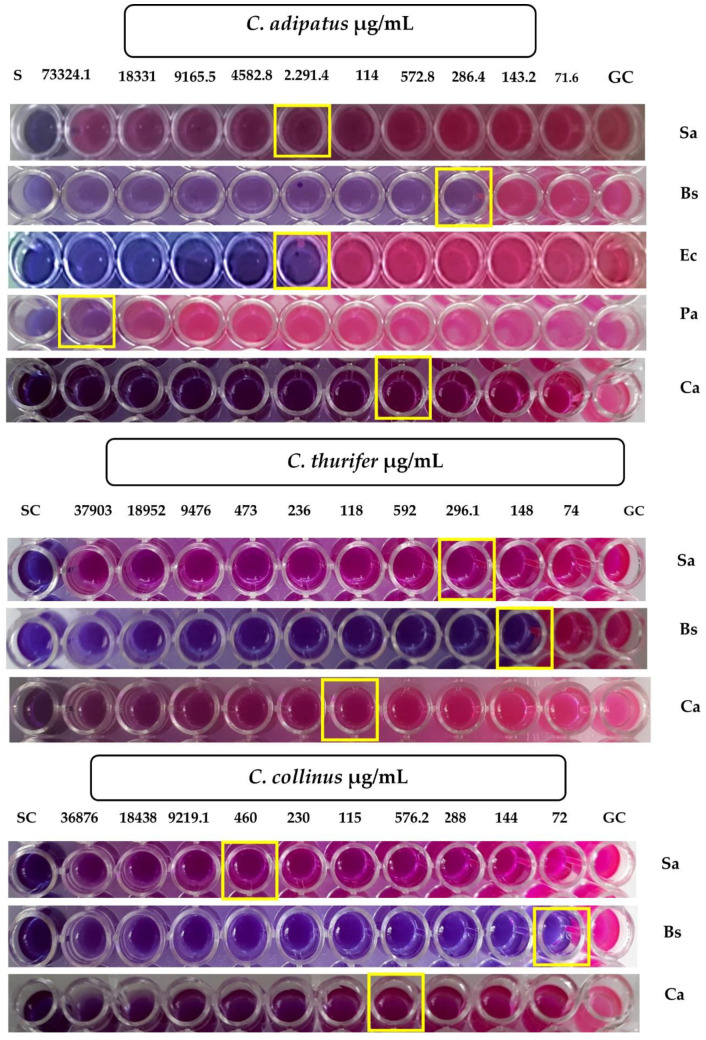
Determination of the minimum inhibitory concentration of the essential oil and standard drugs. SC (Sterility Control), GC (Growth Control), Sa (*S. aureus*), Bs (*B. subtilis*), Ec (*E. coli*), Pa (*P. aeruginosa)*, Ca (*C. albicans*).

**Table 1 antibiotics-10-01387-t001:** Chemical composition of *C. adipatus, C. thurifer*, and *C. collinus* essential oil.

N°	Chemicals	*C. adipatus*	*C. thurifer*	*C. collinus*	Chemical Group
		%	%	%	
1	Isobutyl isobutyrate	0.2			Carboxylic acid esters
2	α-Thujene	12.69	1.7	0.17	Monoterpene hydrocarbon
3	α-Pinene	3.64	0.96	0.41	Monoterpene hydrocarbon
4	Sabinene	3.17	1.2	0.31	Monoterpene hydrocarbon
5	β-Pinene	1.55	0.38	0.36	Monoterpene hydrocarbon
6	β-Myrcene	18.34		6.79	Monoterpene hydrocarbon
7	Unknown (C_10_H_16_)		22.38		Monoterpene hydrocarbon
8	α-Phellandrene	8.19	3.3	0.61	Monoterpene hydrocarbon
9	Isoamyl isobutyrate	0.53	0.99		Fatty acid ester
10	α-Terpinene	0.56			Monoterpene hydrocarbon
11	p-Cymene		0.36	0.29	Monoterpene hydrocarbon
12	o-Cymene	1.38			Monoterpene hydrocarbon
13	D-Limonene	10.94	1.94	8.73	Monoterpene hydrocarbon
14	β- Thujene			6.96	Monoterpene hydrocarbon
15	β-Phellandrene	4.94	1.44		Monoterpene hydrocarbon
16	Eucalyptol	0.7	0.51		Oxygenated monoterpenes
17	γ-Terpinene		0.88	2.1	Monoterpene hydrocarbon
18	cis-Ocimene	0.43		0.35	Monoterpene hydrocarbon
19	τ−Τerpinene	3.21			Monoterpene hydrocarbon
20	β-Terpineol	0.44			Alcohol monoterpene
21	Terpinolene	0.16		0.17	Monoterpene hydrocarbon
22	β-Linalool	0.4			Oxygenated monoterpenes
23	1,6-Dimethylhepta-1,3,5-triene	0.47			Alkene
24	4-Terpineol	0.74			Oxygenated monoterpenes
25	2-Decanone	0.17			ketone
26	α-Terpineol	0.12			Oxygenated monoterpenes
27	2-Undecanone	2.16		0.72	ketone
28	2-Dodecanone	4.56			ketone
29	2-Carene	0.56			Monoterpene hydrocarbon
30	1,5,5-Trimethyl-6-methylene-cyclohexene			0.19	Monoterpene hydrocarbon
31	Copaene		0.44	0.28	Monoterpene hydrocarbon
32	Unknown I (C_15_H_24_)	0.15		0.23	Sesquiterpene hydrocarbon
33	1-Ethenyl-1-methyl-2,4-bis-(1-methylethenyl)-cyclohexane		0.34		Sesquiterpene hydrocarbon
34	Unknown II (C_15_H_24_)			0.23	Sesquiterpene hydrocarbon
35	β-Elemene	6.47	11.87	6.7	Sesquiterpene hydrocarbon
36	2-Ethyl-1,3-dimethyl-Benzene	0.24			Benzene
37	Cyperene		0.69	1.03	Sesquiterpene hydrocarbon
38	β-Caryophyllene	0.28	1.95	44.7	Sesquiterpene hydrocarbon
39	δ−Εlemene	0.71			Sesquiterpene hydrocarbon
40	Geranyl acetone	0.16		0.59	Monoterpene ketone
41	Unknown III (C_15_H_24_)		0.74		Sesquiterpene hydrocarbon
42	Aromadendrene		1.25	0.23	Sesquiterpene hydrocarbon
43	α-Caryophyllene		0.3	3.73	Sesquiterpene hydrocarbon
44	Valencene			0.53	Sesquiterpene hydrocarbon
45	α-Elemene		0.52		Sesquiterpene hydrocarbon
46	Unknown IV (C_15_H_24_)		1.43		Sesquiterpene hydrocarbon
47	α-Curcumene			0.39	Sesquiterpene hydrocarbon
48	α-Muurolene	0.45			Sesquiterpene hydrocarbon
49	Unknown V (C_15_H_24_)		0.59		Sesquiterpene hydrocarbon
50	Germacrene D	4.78	10.22	2.51	Sesquiterpene hydrocarbon
51	α-Selinene	0.39		0.21	Sesquiterpene hydrocarbon
52	Eremophilene		0.67		Sesquiterpene hydrocarbon
53	β-Cubebene			0.22	Sesquiterpene hydrocarbon
54	Bicyclogermacrene	1.24		3.88	Sesquiterpene hydrocarbon
55	Geranil isobutyrate	0.41			Carboxylic ester monoterpene
56	Unknown I (C_15_H_26_O)		3.73		Sesquiterpene hydrocarbon
57	Unknown II (C_15_H_26_O)		1.62		Sesquiterpene hydrocarbon
58	δ-Cadinene	0.48	1.26	1.96	Sesquiterpene hydrocarbon
59	α-Muurolene			0.51	Sesquiterpene hydrocarbon
60	Unknown VII (C_15_H_24_)	0.31		0.76	Sesquiterpene hydrocarbon
61	Elemol	1.61			Oxygenated sesquiterpene
62	Unknown VIII (C_15_H_24_)	0.69			Sesquiterpene hydrocarbon
63	(-)-Spatulenol			0.58	Oxygenated sesquiterpene
64	Unknown (C_15_H_24_O)			0.89	n.d.
65	Unknown III (C_15_H_26_O)	0.51	0.64		n.d.
66	Unknown (C_15_H_22_O)		0.31		n.d.
67	β-Maaliene	0.11			Sesquiterpene hydrocarbon
68	Unknown (C_9_H_14_O)		1.15		n.d.
69	τ-Muurolene			0.15	Sesquiterpene hydrocarbon
70	Unknown IV (C_15_H_26_O)	0.1	21.8		n.d.
71	τ-Muurolol	0.16		0.34	Oxygenated sesquiterpene
72	α-Cadinol			0.73	Oxygenated sesquiterpene
73	Unknown V (C_15_H_26_O)	0.38	0.28		n.d.
74	Heptadecane	0.12	0.26	0.49	Alkane
Total of identified compounds (%)		100.00	100.00	100.00	

n.d.; not determined.

**Table 2 antibiotics-10-01387-t002:** Comparative table of the antimicrobial activity of essential oils and standards by the agar diffusion method.

Microorganism	Zone of Inhibition (mm)										
	Essential Oil									Positive Control	
	*Croton adipatus*			*Croton thurifer*			*Croton collinus*			CIP [0.05 mg/mL]	KET [0.2 mg/mL]
	10%	50%	95%	10%	50%	95%	10%	50%	95%		
Sa	13.13 ± 0.44 *	20 ± 0 *	14.2 ± 0.7 *	11.47 ± 0.5 *	12 ± 0 *	14.07 ± 0.41 *	10.53 ± 0.55 *	11.03 ± 0.52 *	12 ± 0 *	39 ± 0	n.d.
Bs	13.1 ± 1.32 *	17.13 ± 0.34 *	25 ± 0 *	11.5 ± 0 *	12.53 ± 0.46 *	12.17 ± 1.25 *	12.6 ± 0.8 *	14.53 ± 0.4 *	15 ± 0 *	52.3 ± 1.1	n.d.
Ec	14.03 ± 0.41	16 ± 0	20.03 ± 0.29 *	10 ± 0 *	12.03 ± 0.48 *	11.53 ± 0.5 *	11 ± 0 *	11.57 ± 0.5 *	10.6 ± 1.63 *	15 ± 0	n.d.
Pa	11.27 ± 1.02 *	11.53 ± 0.5 *	13.57 ± 0.85 *	10 ± 0 *	12.5 ± 0 *	11 ± 0 *	11.63 ± 1.31 *	12.7 ± 0.79 *	11.03 ± 0.52 *	34 ± 0	n.d.
Ca	19.67 ± 1.47 ^#^	12.5 ± 0 ^#^	11 ± 0^#^	15.03 ± 0.38^#^	11 ± 0 ^#^	10 ± 0 ^#^	15 ± 0 ^#^	10.53 ± 0.55 ^#^	10 ± 0 ^#^	n.d.	35 ± 0

CIP: Ciprofloxacin; KET: ketoconazole; Sa: *S. aureus*; Bs: *B. subtilis*; Ec: *E. coli*; Pa: *P. aeruginosa*; Ca: *C. albicans*; n.d.: non-determined. * *P* < 0.05; Test Dunnet, mean values were compared to Ciprofloxacin group. ^#^
*P* < 0.05; Test Dunnet, mean values were compared to Ketoconazole group.

**Table 3 antibiotics-10-01387-t003:** Comparative table of the minimum inhibitory concentration (MIC) of the essential oils and standards by the colorimetric microdilution method.

Microorganism	Minimum Inhibitory Concentration (MIC)				
	Essential Oil (µg/mL)			Positive Control (µg/mL)	
	*Croton adipatus*	*Croton thurifer*	*Croton collinus*	Ciprofloxacin	Ketoconazole
Sa	>1000	296.1 ± 0	>1000	0.5 ± 0	n.d.
Bs	286.4 ± 0	148 ± 0	72 ± 0	0.13 ± 0	n.d.
Ec	>1000	>1000	>1000	4 ± 0	n.d.
Pa	>1000	>1000	>1000	32 ± 0	n.d.
Ca	572.8 ± 0	>1000	576.2 ± 0	n.d.	0.03 ± 0

Sa: *S. aureus*; Bs: *B. subtilis*; Ec: *E. coli*; Pa: *P. aeruginosa*; Ca: *C. albicans*; n.d.: non-determined.

## Data Availability

The data that support the findings of this study are available from the corresponding author upon reasonable request.

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
