# Peer review of "Chemical Profile of the Volatile Constituents and Antimicrobial Activity of the Essential Oils from Croton adipatus, Croton thurifer, and Croton collinus"

_antibiotics, 2021, doi:10.3390/antibiotics10111387_

Round 1

Reviewer 1 Report

Title: Antibacterial performance of terpenoids from the Australian plant Eremophila lucida

Antibiotics, MDPI

In this manuscript, authors determine the volatile phytochemical constituents and bactericidal function of 3 Croton species. Their work is not novel but adds to previous knowledge on essential oils. Their methodology and experiment are scientifically sound. Introduction as well as Materials and Methods read well but the Results section is difficult to follow. Authors mention way too many plant species, pathogens and numbers from their own and other studies. They should instead organize their data and assist the reader. There are standard ways to organize datasets much larger than those of this study. Specific examples and suggestions follow in major comments.

Furthermore, effective concentration in agar tests and MICs in microdilution test are way too high. In such concentrations, most essential oils would be lethal to several bacterial species. I believe that additional experiments, in which authors would determine specific chemical constituents that are effective against pathogens, are needed. This would make their work less descriptive and add novelty to meet standards for publication in this Journal.

Major comments:

Tables 1, 2 and 3:  Authors present specific chemical constituents; in respected text in the Result sections they categorized the results in chemical roups. I suggest they add such grouping (monoterpenes, sesquiterpenes etc.) in Table so that reader can correlate text with Tables.

Section 2.1: Since many Constituents are common in all 3 examined species but % composition varies, perhaps authors should develop Venn diagrams, heatmaps or abundance diagrams to correlate findings of the 3 examined species with each other and other species from previous studies. In the current manuscript, the reader has to do all the work to compare profiles.

Section 2.2: Again, the reader has to do all the work and compare current to previous findings mentioned (for example in Lines 157-166). Authors mention “similar results” in previous work and in terms of inhibition zones, but they do not mention working concentrations in such results.

In summary, I got frustrated and could not follow all the plant species and pathogen names and numbers. Authors should organize their data, possibly by integrating data of others in Table 4 next to each plant species (columns) and pathogens (rows) or generate a similar table for other species from previous work in a way that one can make comparisons easily.

Figure 1: Quality and format are not in Journal standards. Authors should split the 3 panels (Ca on top, Ct middle and Cc bottom). Photos should not be distorted. Keep original Length-width ratio. White fonts always work best in black background.

Figure 2: Same problem as in Fig. 1. Please format all panel as the Cipro panel. On the top of each panel mention μg/mL concentration; μL/mL is a relative number and does not mean much.

Lines 132-139: Information provided should be correlated with findings in this study (see general comments above)

Minor comments:

Line 56: Are those chemical defenses correlated to their antimicrobial properties. If yes, please provide examples. If not, omit text from Introduction.

Lines 59-74: Please provide information about previous work in chemical composition of Croton species.

Author Response

Dear Reviewer, 1

Thank you for all the comments and suggestions to improve or manuscript.

According to your comments, we replied point by point and were highlighted in yellow in the manuscript submitted to Antibiotics Journal.

In this manuscript, authors determine the volatile phytochemical constituents and bactericidal function of 3 Croton species. Their work is not novel but adds to previous knowledge on essential oils. Their methodology and experiment are scientifically sound. Introduction as well as Materials and Methods read well but the Results section is difficult to follow. Authors mention way too many plant species, pathogens and numbers from their own and other studies. They should instead organize their data and assist the reader. There are standard ways to organize datasets much larger than those of this study. Specific examples and suggestions follow in major comments.

Furthermore, effective concentration in agar tests and MICs in microdilution test are way too high. In such concentrations, most essential oils would be lethal to several bacterial species. I believe that additional experiments, in which authors would determine specific chemical constituents that are effective against pathogens, are needed. This would make their work less descriptive and add novelty to meet standards for publication in this Journal.

Major comments:

Tables 1, 2 and 3:  Authors present specific chemical constituents; in respected text in the Result sections, they categorized the results in chemical groups. I suggest they add such grouping (monoterpenes, sesquiterpenes etc.) in Table so that reader can correlate text with Tables.

R1: Thank you for your suggestions, we modified our results, and they were grouped in a only table to improve our manuscript.

Section 2.1: Since many Constituents are common in all 3 examined species but % composition varies, perhaps authors should develop Venn diagrams, heatmaps or abundance diagrams to correlate findings of the 3 examined species with each other and other species from previous studies. In the current manuscript, the reader has to do all the work to compare profiles.

R2: Thank you, a Venn diagram was included. We discussed our findings with other results of Croton species.

Section 2.2: Again, the reader has to do all the work and compare current to previous findings mentioned (for example in Lines 157-166). Authors mention “similar results” in previous work and in terms of inhibition zones, but they do not mention working concentrations in such results.

R3. Thank you for your comments, we modified this section adding concentrations and its main results on each bacterium.

In summary, I got frustrated and could not follow all the plant species and pathogen names and numbers. Authors should organize their data, possibly by integrating data of others in Table 4 next to each plant species (columns) and pathogens (rows) or generate a similar table for other species from previous work in a way that one can make comparisons easily.

R4: Thank you for your comments, we modified the presentation of our results, and they were ordered in paragraphs for each bacterium and fungi with other findings found of Croton. We tried to improve it considering that we are working with three essential oils.

Figure 1: Quality and format are not in Journal standards. Authors should split the 3 panels (Ca on top, Ct middle and Cc bottom). Photos should not be distorted. Keep original Length-width ratio. White fonts always work best in black background.

R4: Thank you for your comments, we modified, and a new figure was generated to improve our presentation in the manuscript.

Figure 2: Same problem as in Fig. 1. Please format all panel as the Cipro panel. On the top of each panel mention μg/mL concentration; μL/mL is a relative number and does not mean much.

R5: Thank you for your comments, we modified the figure in microdilution test, we only showed the results of three essential oils, Standard drug results were included as supplementary material.

Lines 132-139: Information provided should be correlated with findings in this study (see general comments above)

R6: Thank you for your comments, in effect our results were correlated with the main phytochemicals determined in the GC-MS analysis. We added two new paragraphs as follows:

In Agar diffusion:

Results for this type of preliminary antimicrobial test showed a certain degree of variability, which could be explained by different factors that influence the solubility of the essential oils in the medium to be diffused, the polarity of the essential oil, and the molecular size of the components that comprise the essential oil, among others; therefore, they would not necessarily correlate with the results of other specific tests such as microdilution or dilution in broth. Regarding the biochemical mechanisms involved in the antimicrobial activity of essential oils, some compounds like Ñ€-Cymene at 2500 µg/mL have showed alterations in the membrane potential of S. aureus, and also γ-terpinene at 2500 and 3400 µg/mL [40]. β-caryophyllene, the major component in C. collinus was able to alter membrane permeability and integrity of B. cereus, leading possibly to cell death [41]. This finding is related to our results against B. subtilis, which only was selective for this strain (Table 2 and Table 3). Limonene, the third abundant component in C. adipatus is linked to affect the respiratory metabolism of S. aureus [42]. Otherwise, terpenes are also related to disorders of mitochondrial membrane in Candida species but with low antifungal activi-ty. Moreover, oxygenated sesquiterpenes demonstrated a negative effect on Saccharomyces and Candida strains such as caryophyllene oxide, humulene epoxide, spathulenol, and α-Muurolol in high concentrations [43].

In microdilution test:

Based on our results, we might suppose that the three essential oils had a selective effect against a specific bacterium and on C. albicans. Each metabolite determined could be acting by different mechanisms and according to our chemical analysis each Croton spe-cies showed a different phytochemical marker, which contribute to the antimicrobial activity. Hydrocarbon monoterpenes were the most abundant chemical groups in C. adipatus. However, in C. thurifer and C. collinus were the sesquiterpenes, which might be linked to this difference in the results showed in Table 2 and Table 3. Therefore, the volatile chemicals would be the responsible for the antimicrobial effect on S. aureus, B. subtilis, and C. albicans and not the major phytochemical determined in the GC-MS analysis. Some studies reported that terpenes do not constitute a high inherent antimicrobial effect i.e., some terpenes like p-cymene, the major compound in thyme, had not any effect against several Gram-negative bacteria. Also, other chemicals, which were found in Croton essential oils such as α-pinene, limonene, δ-3-carene, sabinene, α-pinene, β-pinene, and α-terpinene revealed no or low effect against different pathogen bacteria Gram-negative and Gram-Positive. Additionally, α-pinene, β-pinene, p-cymene, β-myrcene, β-caryophyllene, limonene, and γ-terpinene had low or absent activity against Escherichia coli, Staphylococcus aureus, and Bacillus cereus and p-Cymene and γ-terpinene were ineffective against Saccharomyces cerevisiae [40]. Thus, these reports in vitro indicate that single terpene compounds are inefficient as antimicrobials which support our study to under-stand the relationship between the chemical composition of each Croton species with the antimicrobial activity.

Minor comments:

Line 56: Are those chemical defenses correlated to their antimicrobial properties. If yes, please provide examples. If not, omit text from Introduction.

R: Thank you for your observation, we modified this sentence and was excluded of the text.

Lines 59-74: Please provide information about previous work in chemical composition of Croton species.

R: Thank you for you observation, this was added in the introduction:

a) Essential oils from Croton species have presented several volatile constituents as finger-print, which might be related to its antimicrobial activity, i.e., C. ceanothifolius showed Bi-cyclogermacrene (26.3%), Germacrene D (14.7%), and E-caryophyllene (11.7%) as major components [11], in C. heliotropiifolius were found β-Caryophyllene as the major constituent followed by Bicyclogermacrene, Germacrene-D, Limonene, and 1,8-Cineole [12], in C. argyrophyllus had bicyclogermacrene (38.42 %), (Z)-caryophyllene (14.06 %), epi-longipinanol (9.78 %), and germacrene B (9.1 %) [13]. Additionally in C. zenhtneri, C. nepetaefolius, C. argyrophyllus, C. sonderianus were trans-Anethole (94.09%), Methyl eugenol (48.47%), α-Pineno (20.96%), and β-Phellandrene (18.21%) respectively [14], finally in C. heliotropiifolius were identified β-caryophyllene (35.82%), bicyclogermacrene (19.98%), and germacrene-D (11.85%), in C. pulegiodorus were identified β-caryophyllene (20.96%), bicyclogermacrene (16.89%), germacrene-D (10.55%), τ-cadinol (4.56%), and β-copaen-4-α-ol (4.35%) [15].

b) In regard to our Croton species, we could not find articles showing their chemical compositions. Therefor our report was the first study on these Peruvian species.

Reviewer 2 Report

  1. Results and discussion

 2.1. Phytochemical screening of the volatile components of the essential oils from Croton species.

-The tables with identified compounds do not bring nether the retention time nor Adams Index (AI) references, traditionally reported in this type of analysis. It is not necessary repeat the name and concentrations in the text, but pointed only those with highest concentration, comparing with the regions, as example.

-What were total (%) of all volatile compounds (VCs) in each studied Croton species?

- α-Pinene but not 1R-α-Pinene compound in tables 1 and 3;

2.2 Antimicrobial activity of the essential oils of Croton species

-Standardize: SA or Sa?

2.2.2. Colorimetric microdilution method

Line 226: C. borarium (MIC=1.25 µL/mL or 1151 µg/mL)

  1. Materials and Methods

This section is number 3!

3.1 Collection of the plant material

Line 316: had all leaves from croton species collected in June 2017? Rainy or dry season?

3.2. Extraction of essential oils of Croton species

Line 324: what was the solvent used in the Clevenger apparatus?

3.3. Chemical analysis by Gas Chromatography coupled to Mass Spectrometry (GC-MS) of the essential oils of Croton species

- retention index (RI) calculated, and RI reference are not present in tables 1-3 ;

-What were the used standard samples?

-Was the analysis done in triplicate?

Reviewer 3 Report

In this work Juana Liz Cucho-Medrano et al., have tested the Essential Oils from Croton adipatus, Croton thurifer, and Croton collinus concerning constitution and antimicrobial activity. This is a nice work including selection, collection, treatment and oil preparation, and the results are  interesting although a clear conclusion is missing. I suggest to be accepted after addition of a 3 number paragraph entitled Conclusions (actually 3 paragraph is missing). There to be clear which one fraction has the higher antimicrobial activity, the constituents of the fraction and the next steps to be identified the responsible active componets of antimicrobial activity

Author Response

Dear Reviewer, 

Thank you for all the comments and suggestions to improve or manuscript.

According to your comments, we replied point by point and were highlighted in yellow in the manuscript submitted to Antibiotics Journal.

In this work Juana Liz Cucho-Medrano et al., have tested the Essential Oils from Croton adipatus, Croton thurifer, and Croton collinus concerning constitution and antimicrobial activity. This is a nice work including selection, collection, treatment and oil preparation, and the results are  interesting although a clear conclusion is missing. I suggest to be accepted after addition of a 3 number paragraph entitled Conclusions (actually 3 paragraph is missing). There to be clear which one fraction has the higher antimicrobial activity, the constituents of the fraction and the next steps to be identified the responsible active components of antimicrobial activity

R1: Thank you so much for your comments and suggestions. We added a section of conclusions.